# Erector Spinae Plane Block and Chronic Pain: An Updated Review and Possible Future Directions

**DOI:** 10.3390/biology12081073

**Published:** 2023-08-01

**Authors:** Alessandro De Cassai, Federico Geraldini, Ulderico Freo, Annalisa Boscolo, Tommaso Pettenuzzo, Francesco Zarantonello, Nicolò Sella, Serkan Tulgar, Veronica Busetto, Sebastiano Negro, Marina Munari, Paolo Navalesi

**Affiliations:** 1Anesthesia and Intensive Care Unit, University-Hospital of Padua, 35128 Padua, Italy; 2Department of Medicine, University of Padua, 35122 Padua, Italy; 3Thoracic Surgery and Lung Transplant Unit, Department of Cardiac, Thoracic, Vascular Sciences and Public Health, University of Padua, 35128 Padua, Italy; 4Department of Anesthesiology and Reanimation, Training and Research Hospital of Samsun, Faculty of Medicine, University of Samsun, 55000 Samsun, Turkey; 5Cardiac Surgery Intensive Care Unit, University Hospital of Padua, 35128 Padua, Italy; 6Anesthesia and Intensive Care 2, Istituto Oncologico Veneto IRCCS, 35128 Padua, Italy; 7Sant’Antonio Anesthesia and Intensive Care Unit, University-Hospital of Padua, 35128 Padua, Italy

**Keywords:** anesthesia, conduction anesthesia, chronic pain, review

## Abstract

**Simple Summary:**

Erector spinae plane block is a technique used by anesthesiologists and pain physicians. It was introduced in 2016 and consists of an injection of local anesthetic between a vertebra and its corresponding muscle (erector spinae). It provides diffuse somatic and visceral analgesia useful for both surgery and pain therapy. In our paper we overview chronic pain, fascial blocks and erector spinae plane blocks. We give an overview of the erector spinae plane block technique, complications and possible use in chronic pain settings, highlighting the current evidence with a final overview of possible future directions of research.

**Abstract:**

Chronic pain is a common, pervasive, and often disabling medical condition that affects millions of people worldwide. According to the Global Burden of Disease survey, painful chronic conditions are causing the largest numbers of years lived with disability worldwide. In America, more than one in five adults experiences chronic pain. Erector spinae plane block is a novel regional anesthesia technique used to provide analgesia with multiple possible uses and a relatively low learning curve and complication rate. Here, we review the erector spinae plane block rationale, mechanism of action and possible complications, and discuss its potential use for chronic pain with possible future directions for research

## 1. Chronic Pain

Chronic pain is a common, pervasive, and often disabling medical condition that affects millions of people worldwide. According to the Global Burden of Disease survey, painful chronic conditions are causing the largest numbers of years lived with disability worldwide [1]. In America, more than one in five adults experiences chronic pain [2]. Chronic pain could be defined as a pain persisting for more than 3 months [3]; it can range from mild to severe intensity and originate from different injuries or illnesses of body tissues and/or of the somatosensory nervous systems.

One of the most challenging aspects of chronic pain is that, in contrast to acute pain, it often does not respond to conventional pain management techniques alone, such as medications or physical therapy [4]. This can lead to frustrating feelings of hopelessness and despair, as individuals struggle to find relief from their suffering. Chronic pain can also have a profound impact on a person′s quality of life, affecting several domains of a person’s life such as physical (disability, muscular strength and endurance, performance in activities of daily living, and body composition), psychological (kinesiophobia, fear-avoidance, pain catastrophizing, pain self-efficacy, depression, anxiety, and sleep quality), social (social functioning and work absenteeism), and health-related quality-of-life [5].

Despite the significant impact that chronic pain can have on a person′s life, it is often misunderstood and undertreated, especially in elderly, non-communicating or fragile patients (e.g., patients with dementia) [6]. Moreover, many people with chronic pain are hesitant to seek help due to stigma or fear of being dismissed by medical professionals without a proper medical intervention [7], while other patients suffering from this debilitating condition may not have access to adequate healthcare or resources to manage their pain.

However, there are many effective treatment options available for chronic pain, including medications, regional anesthesia, physical therapy, psychological interventions, and alternative therapies [8,9]. By understanding the complex nature of chronic pain and working with healthcare professionals to find a personalized treatment plan, individuals with chronic pain can regain control over their lives and find relief from their suffering [10].

## 2. Mechanism of Chronic Pain

The mechanism of chronic pain is complex and involves multiple physiological and psychological factors [11]. According to the IASP terminology, pain can be categorized into nociceptive, neuropathic (central or peripheral) or nociplastic pain, based on whether it results from a disease affecting the body tissues, the somatosensory nervous system, or from a heightened sensitivity without a demonstrable lesion [3]. Although it is not included in the IASP terminology, “mixed pain” is increasingly used by pain clinicians when referring to a complex and simultaneous overlap of nociceptive, neuropathic and nociplastic mechanisms in specific clinical conditions and frequently in chronic pain patients.

Chronic pain is associated with peripheral and central sensitization occurring in the peripheral nociceptors and dorsal root ganglia, the spinal cord and cognitive and emotional brain areas [12,13].

Peripheral sensitization is a condition where the pain threshold decreases, leading to an increase in pain sensitivity [13]. Chemical mediators released by nociceptors and non-neuronal cells of inflammation trigger peripheral sensitization, causing changes in the environment surrounding nociceptors in the affected area [13]. The release of signaling molecules such as proteins, ATP, prostaglandins, growth factors, cytokines, and neuropeptides leads to the expression of voltage-gated sodium channels and a reduction in the discharge threshold of dorsal root ganglia neurons. This causes nociceptive nerve fibers to become more responsive to sensory stimuli, leading to increased firing of action potentials and the release of transmitters in the dorsal horn of the spinal cord. Peripheral sensitization is commonly seen in rheumatic, surgical and trauma pain, and can lead to central sensitization over time [14,15,16].

Sensitization can cause persistent pain even after the injury or illness has healed. Pain receptors can become activated chronically, causing heightened sensitivity and excitability in the central nervous system. The wide dynamic range of neurons respond to different pain types and their responses increase with stimulus frequency, causing the “wind up” phenomenon [17]. Maladaptive neuroplastic changes in neurons can lead to chronic neuropathic pain after a peripheral nerve is lesioned [18]. An increase in NMDA-type glutamate receptors and neuropeptide receptors can cause a rapid elevation in intracellular calcium levels and alter the activity of nociceptive circuits; this amplifies the pain signal, causing even innocuous stimuli to be perceived as extremely painful (allodynia) [19]. Understanding the mechanisms of pain chronification is critical for the development of effective pain management strategies [14,15].

Psychological factors can also play a significant role in chronic pain, and a recently developed evidence-based classification highlights the role of social and psychological factors in chronic pain development [20,21].

Finally, genetics can also play a role in the development of chronic pain and in the variability in the perception of pain. Family studies show a strong familial aggregation of some chronic pain diseases, and recent evidence suggests a role for polymorphisms of genes in the serotoninergic, dopaminergic and catecholaminergic systems [22].

Because emerging evidence in the last two decades suggest that chronic pain is associated with a lower quality of life and shortened life expectancy [23], chronic pain is now viewed as a disease on its own, regardless of the underlying disease or injury.

Opioids are indispensable for the treatment of severe pain [24]. However, the long-term (over)prescribing of high doses of opioids, mostly for chronic pain, led to unacceptable rates of complications and deaths with a negative socio-economic impact and decline of life expectancy in North America: the opioid epidemic. Hence, the value of opioids has been questioned and alternative non-opioid analgesia techniques have been put forth for postoperative, trauma and especially chronic pain conditions.

## 3. Regional Anesthesia and Chronic Pain

Regional anesthesia, which involves the injection of local anesthetics near nerves or nerve roots, can play a significant role in the management of chronic pain [25]. This technique can be used to block pain signals from specific areas of the body, providing targeted pain relief without the side effects associated with systemic pain medications. Possible advantages of regional anesthesia, such as epidural or nerve blocks, over systemic medications alone are: targeted pain relief to the specific area being treated; a lower dose of medication to be used, minimizing risk of side effects associated with systemic medications; a longer-lasting pain relief compared to systemic medications, which may require frequent dosing, when a continuous peripheral nerve block, for example, can provide pain relief for several days or even weeks [26], and its effects on pain relief usually outlast the half-life of local anesthetic used [27], allowing individuals with chronic pain to engage in daily activities and physical therapy without experiencing excessive pain.

Regional anesthesia for chronic pain is particularly indicated for patients with pathology concerning dermatomal pain distribution patterns and without depression, opioid use, high baseline disability, and pain scores [28]. For example, an epidural block can be used to relieve pain caused by a herniated disc or a nerve block could be used to provide pain relief in case of nerve injury.

Regional anesthesia can also be used to treat nociceptive and neuropathic chronic pain caused by inflammation or injury to specific joints or tissues or of peripheral nerves. For example, an epidural steroid injection can be used to reduce inflammation and relieve pain in the lower back or neck. Similarly, a joint injection or troncular anesthesia can be used to provide pain relief in the knee, shoulder or hip.

However, it is important to note that regional anesthesia is not a one-size-fits-all solution for chronic pain. In fact, this is only a piece of the puzzle of chronic pain treatment that is usually based on a comprehensive treatment plan that also includes medications, physical therapy, and behavioral interventions [29].

## 4. Fascial Plane Blocks

Fascial plane blocks are a type of regional anesthesia that target the planes of connective tissue in the body. Although fascial plane blocks, such as the transversus abdominis plane block, were first described by Rafi in 2001 using a landmark-guided technique with loss of resistance [30], the routine use of ultrasound in operating rooms and pain clinics has led to a significant increase in research on the topic. Ultrasound allows anesthesiologists visualize planes and structures that are not always perceptible with the landmark approach. In the past decade, there has been an academic surge in studies on fascial plane blocks, with hundreds of publications each year dedicated to them, and a recent publication highlighting that over a period of six months 69 articles related to pectoserratus or interpectoral nerve blocks (previously known as PECS II and PECS I, respectively) have been published [31]. These blocks are becoming increasingly popular because of their ability to provide effective pain relief while minimizing the risk of systemic side effects and with minimal rate of complications related to these techniques [32]. There are several different types of fascial plane blocks, each targeting a specific fascial plane. Of note, the increase in research on the topic caused heterogeneity in the names and anatomical descriptions of fascial blocks, with the ultimate consequence of possible adverse consequences for education, research and implementation into clinical practice. For this reason, in 2020 an ESRA/ASRA joint consensus [33] produced a standardized nomenclature for abdominal wall, paraspinal, and chest wall regional anesthetic techniques with a total of 20 described techniques (Table 1).

## 5. Erector Spinae Plane Block

The erector spinae plane (ESP) block is a relatively new fascial block that was first described in 2016 by Forero [34]. Despite being a relatively recent development, it has gained widespread popularity in both academic and clinical settings [31]. This block targets the plane between the erector spinae muscle group and the transverse processes of the vertebrae. The erector spinae muscle group is a critical group of muscles that run along the length of the spine, providing stability and movement to the back. This block must be performed under ultrasound guidance in order to avoid unwanted complications such as pneumothorax when performed at thoracic level. It could be performed with a linear high-frequency probe for thoracic vertebrae and with a convex low frequency probe for lumbar vertebrae as their transverse processes are usually further from the skin.

To perform an ESP block, the patient could be positioned in a sitting, lateral or prone position. If the prone position is chosen, it could be helpful to place a pillow or cushion under their abdomen. The ultrasound probe is placed in a longitudinal orientation at the level of the intended block, usually starting at the T4 or T5 level. The probe is then moved laterally until the transverse process of the targeted vertebra is identified, which will appear as a trapezoidal hyperechoic structure on the ultrasound image. This trapezoidal hyperechoic structure is easily differentiated by ribs, as they appear as a rounded hyperechoic structure.

The needle is inserted in-plane with the ultrasound probe, and advanced until it reaches the fascial plane between the erector spinae muscle group and the transverse process (Figure 1 and Figure 2) [35].

The correct placement of the needle can be confirmed by the visualization of local anesthetic spreading between the erector spinae muscle group and the transverse process on the ultrasound image. The correct volume of anesthetic to be injected has not been formally determined. Although a fixed volume of at least 20 mL is used in the literature, recent studies suggest that a higher volume of anesthetic (for example, 30 mL) could be helpful [36].

The ESP block has become increasingly popular due to its ease of administration, fast learning curve, low reported complication rates and its ability to be used in patients with impaired coagulation [37].

## 6. ESP Block Mechanism of Action

The mechanism of action of the ESP block is not fully understood, and several mechanisms have been proposed over the years. It is believed to be multifactorial. Recently, Chin et al. [38] published a comprehensive narrative review regarding the topic.

The efficacy of ESP blocks for pain management remains controversial due to conflicting evidence about how the blocks work. Evidence suggests that local anesthetic can and does spread into the thoracic paravertebral space, which was initially proposed as the primary mechanism of action of the ESP block, with some studies suggesting that an epidural spread is also possible [36]. This is due to the posterior thoracolumbar fascia and inter-transverse connective tissue complex, which is perforated by branches of the dorsal rami and accompanying blood vessels, allowing local anesthetic to track into the paravertebral space gradually. However, it is important to highlight that the ESP block does not produce both the pressure-like chest discomfort or the movement of the pleural line at the ultrasound, and both these signs are often associated with thoracic paravertebral blockade, indicating that the anesthetic seeps slowly rather than rapidly distending the paravertebral space [39]. Cutaneous sensory loss is also not always consistent, calling into question blockade of the ventral rami within the paravertebral space as the underlying mechanism of analgesia [40]. While initial attention was given to its effects on the ventral rami of spinal nerves, it is now clear that the physical spread of local anesthetic associated with the ESP block also affects the dorsal rami, which innervates the spine and paravertebral tissues [41].

The spread of local anesthetic in the lumbar spine is different from that in the thoracic spine due to anatomical differences. However, studies have shown that the injectate can spread to the anterior aspect of the transverse processes and posteromedial border of the psoas muscle, with staining of the spinal nerves in many cases [42]. In the cervical spine, injection at the C6 and C7 level consistently produced staining of the nerve roots that innervate the shoulder girdle [43].

While it has been suggested that the clinical effects of the ESP block are primarily due to an isolated blockade of the lateral cutaneous branches, there is ample clinical evidence that local anesthetic remains around erector spinae muscles and it probably does not reach the lateral cutaneous branches [38].

The injection of large volumes of local anesthetic into fascial plane blocks, such as the ESP block, at doses close to maximum recommended limits may produce plasma concentrations that have systemic analgesic effects [44]; a radiological study investigating ESP spread in volunteers showed that vascular structures largely and rapidly uptake local anesthetics from the ESP plane [36] (Figure 3).

Recently, Fusco et al. [45] proposed two further possible mechanisms for the ESP block’s mechanism of action. Firstly, they suggested that the fascia itself could be a target, as it was demonstrated in previous studies that the fascia is rich in free nerve endings. Secondly, they hypothesized a role of muscle relaxation provided by local anesthetics, providing a preliminary demonstration with elastosonography [45].

While it has been shown that intravenous lidocaine infusions have analgesic benefits in managing acute pain, with mechanisms that involve both neural and non-neural sites of action, those analgesic benefits have not been shown for other commonly used local anesthetics such as ropivacaine, bupivacaine, and levobupivacaine. Systemic lidocaine inhibits excitatory activity of wide dynamic range neurons in the dorsal horn and nociceptive transmission, as well as inhibiting action and potential propagation from A-delta and C-fiber nociceptors [46]. Lidocaine also inhibits several different elements of the inflammatory pathway, contributing to its efficacy in treating conditions such as renal colic and critical limb ischemia [47].

A further, yet speculative, mechanism is the direct action of local anesthetic on a richly innervated organ such as the fascia [48], as such anatomical compartments are richly innervated, especially by proprioceptors and nociceptors [48].

## 7. ESP Block Rate of Complications

Fascial plane blocks also have a lower risk of systemic side effects compared to other types of regional anesthesia. This is because the local anesthetic is injected in a fascial plane, far from important structures such as a plexus or neuraxis. The rate of complication for ESP blocks has been estimated to be as low as two cases every 10,000 patients [32], even if this estimation is only a statistical model speculation based on zero complicationsin the events retrieved; ESP block complications have been reported in case reports, case series and randomized controlled trials. Motor weakness or motor block have been reported [49,50], most commonly observed in cases of high-volume injections or when the block spreads to the spinal cord or nerve roots. Given the anatomical proximity of pleura to the transverse process of the vertebra, it is unsurprising that pneumothorax have been described [51]. Pneumothorax can occur if the needle penetrates the pleura or lung tissue during a thoracic ESP block. This can lead to respiratory distress and requires immediate intervention. To minimize this risk, practitioners should use ultrasound guidance to confirm the needle tip location and avoid inserting the needle too deeply.

The most serious complication of any local anesthetic technique is systemic toxicity (LAST), which can occur if the anesthetic solution is inadvertently injected into a blood vessel or if too much solution is injected. Additionally, practitioners should be aware that the rate of adsorption of local anesthetics from the ESP plane into systemic circulation is relatively fast with Tmax, happening only five minutes after the injection. These findings have been consistently found for both lidocaine [44] and ropivacaine [52]. Until additional data are available, physicians should consider that different rates of adsorption could occur in different groups of patients related to age, gender, weight, etc. Signs of systemic toxicity include central nervous system depression, seizures, cardiovascular collapse, and respiratory arrest [53]. To prevent systemic toxicity, practitioners should use ultrasound guidance to confirm the needle tip location and avoid injecting large volumes of local anesthetic [54]. LAST following an ESP block has been well reported and described in a case series by Tulgar et al. [49].

While hematoma formation is a possible risk, like any needle-based procedure, it is important to highlight that the ESP block is performed far from the neuraxis, in a compressible plane and has been performed in patients with bleeding disorder [55] and in several studies on cardiac surgery [56].

## 8. ESP Block and Chronic Pain

ESP block use has become more common as it was initially indicated and performed for relief of thoracic neuro-pathic pain and has now been used for perioperative analgesia in all surgeries from cervical to knee [49]. However, while the use of the ESP block for different surgeries has been widely explored [57,58], the evidence for ESP blocks for chronic pain has been limited generally to anecdotal reports and to studies of limited quality [59], even if very recent randomized controlled studies have also been published, finally showing a change in this trend. In a controlled study conducted by Guven Kose et al., it was reported that ESP blocks applied in patients with a diagnosis of interscapular myofascial pain syndrome provided very rapid relief in pain and reduced pain intensity and analgesia requirement for a minimum of 6 weeks compared to basal values [58]. Just as there is an ongoing debate about the use and possible uses of the ESP block in perioperative analgesia, the debate continues for the mechanism of action in chronic myofascial pain [58]. Although some clinical studies and observations support the efficacy of the ESP block, we still do not have sufficient data to evaluate efficacy using meta-analysis.

The use of ESP blocks in chronic myofascial pain (cervical and interscapular) was first reported in 2019, and randomized studies have shown that the ESP block is an effective pain relief method both when added to traditional treatment and when used alone [60,61]. However, as the source of pain in myofascial pain syndromes is controversial, the mechanism of this pain relief of ESP blocks and similar fascial plane blocks has not been fully clarified. This relaxation may be due to the local anesthetic/corticosteroid effect on the terminal branches of the dorsal ramus of the spinal nerves, or it may also be due to the separation of the fascial adhesions by hydrodissection [62,63] or a direct effect on fascial nociceptors [45]. A recent retrospective study also showed the potential use of ESP blocks for cancer-related chronic pain, as in a cohort of 110 patients the ESP block was effective in reducing the pain in 53% of the patients [64].

Another potential use for the ESB block in chronic pain is radiculopathies at different levels. With the radiological evaluations, it has been shown that the injectate passes into the epidural space through the transforaminal route in the ESP block applied with high volume from the lumbar region [65]. In addition, with cadaver studies and case reports, it has been reported that epidural spread occurs in cervical and sacral radiculopathies with an appropriate level of ESP block—and sometimes even to the opposite side—and that ESP blocks can have an epidural injection-like effect [66,67]. Although it cannot go beyond anecdotal reports at the moment, the use of ESP remains a possible option due to its relative ease and safety, with indications that it can aid with conditions such as chronic headache, shoulder pain, etc. The sacral ESP block (or multifidus plane block) is a plane block defined by Tulgar et al., targeting the posterior branches of the sacral nerves. This technique could be a new path for sacral radiculopathies as well as entrapments of cluneal nerves and other gluteal pain syndromes [68].

Although the first case of its use is thoracic neuropathic pain, which is a type of chronic pain, there are still no randomized studies with high evidence on the use of the ESP block in various types of chronic pain, unlike the studies demonstrating its perioperative analgesic effect. The ESP block in chronic pain is still a mystery for us; its mechanism awaits clarification and its effectiveness and duration of efficacy remain to be revealed, as do its advantages over conventional techniques.

The ESP block was introduced less than a decade ago, and therefore ongoing studies are being conducted to verify its effectiveness. Undoubtedly, the ESP block has proven to be a safe and effective approach when compared to placebos or no intervention. However, there is still much to explore in terms of its efficacy when compared to other techniques.

A recent study published in the *British Journal of Anaesthesia* showed that the ESP block was as effective as a paravertebral block in reducing both the acute postsurgical pain and the chronic postsurgical pain at three months after minimally invasive thoracic surgery [69]. While it is interesting that the ESP block has been introduced and endorsed by Regional Anaesthesia UK as a plan “A” block among other six regional blocks whichdesignated seven regional blocks covering the anatomical locations commonly encountered in surgery and acute pain [70], the current evidence supporting its routine use for chronic pain is still lacking. For this reason, a call to action of future studies comparing ESP block with other regional techniques, especially in different chronic pain settings, is of paramount importance.

## 9. Future Direction

As the use of the ESP block for chronic pain management continues to gain attention and popularity, it is likely that future research will focus on several areas related to this technique.

One area of interest is the investigation of optimal doses, volumes, and concentrations of local anesthetics and adjuvants used in the ESP block. There is still much to learn about the mechanisms of action of the ESP block, and researchers may explore the effects of different local anesthetics and adjuvants on pain relief, the duration of block and potential complications.

Another potential area of research is the use of the ESP block for chronic pain in specific conditions or populations. For example, future studies may examine the efficacy of ESP blocks in patients with chronic pain related to spinal stenosis or degenerative disc disease. Additionally, research may focus on the use of the ESP block in specific populations such as elderly patients, pregnant women and patients with comorbidities.

In the next ten years, researchers may also explore the use of the ESP block in combination with other treatment modalities such as physical therapy, cognitive behavioral therapy and pharmacological agents. This approach may help to optimize pain management in patients with chronic pain.

Furthermore, as the ESP block is a relatively new technique, there is a need for standardized protocols and guidelines for performing the block. Future studies may investigate the safety and effectiveness of different techniques for performing the ESP block, and provide recommendations for optimal patient positioning, needle placement, and injection technique.

Finally, as the use of the ESP block for chronic pain management becomes more widespread, it will be important for future research to focus on the long-term outcomes of the technique. Studies may examine the durability of pain relief, the need for repeat injections and the potential for adverse effects, such as nerve injury or infection.

Overall, there is a promising future for research on the use of the erector spinae plane block for chronic pain management. Future studies may explore the optimal use of this technique in different populations, the combination of ESP blocks with other therapies and the long-term outcomes of the technique.

## 10. Conclusions

The ESP block is a novel technique with a low rate of complications, at the same time being easy to perform and having multiple possible uses in chronic pain. However, we need further evidence to fully understand its real role and the most suitable use for this block in chronic pain settings.

## Figures and Tables

**Figure 1 biology-12-01073-f001:**
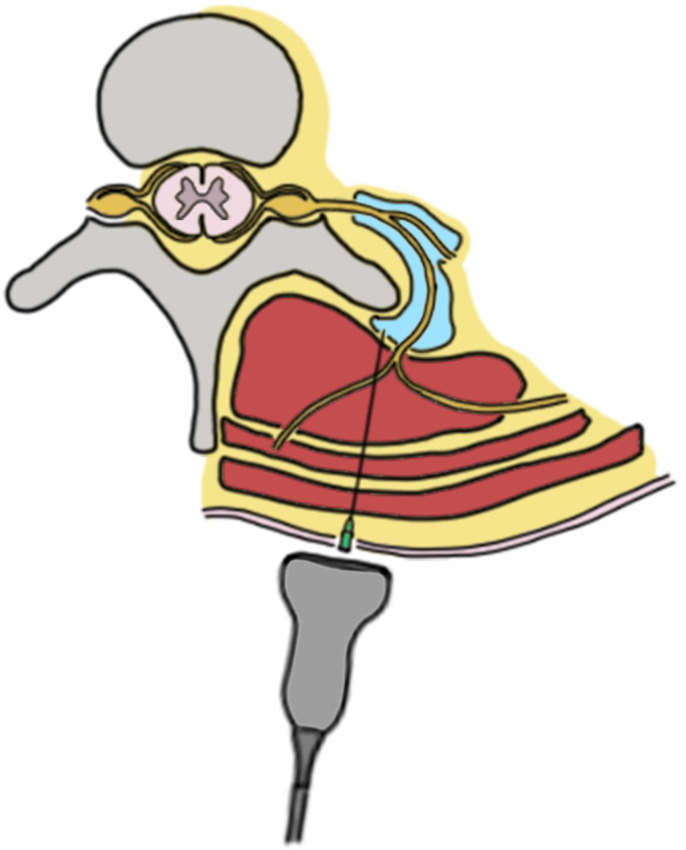
Graphic representation of ESP block. Under ultrasound guidance, the needle is advanced between the erector spinae muscle group and transverse process; after contact with the bone, the local anesthetic can be injected.

**Figure 2 biology-12-01073-f002:**
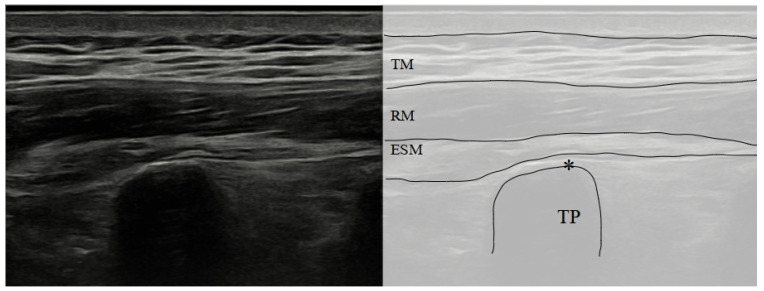
Sonoanatomy of ESP block. TM: Trapezius Muscle; RM: Rhomboid Muscle; ESM: erector spinae group muscle; TP: transverse process; *: target.

**Figure 3 biology-12-01073-f003:**
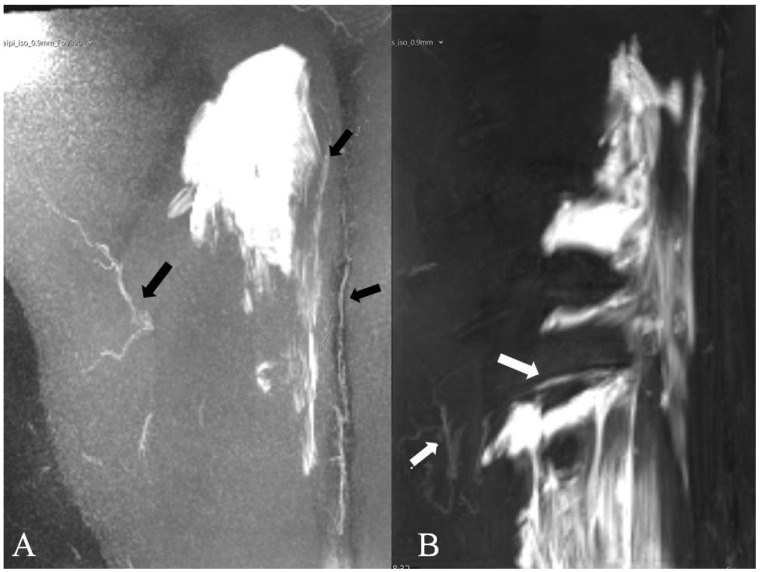
Coronal view of magnetic resonance imaging of two patients receiving ESP block with a 30 mL volume (gadolinium). Vascular uptake is shown with black arrows in panel (**A**) and with white arrows in panel (**B**). The images are unpublished material from a previously published study [36]; images are reproduced by courtesy of Marie Sørenstua.

**Table 1 biology-12-01073-t001:** Fascial blocks adopted nomenclature.

Site	Number	Name
Abdominal wall	1	Rectus sheath block
	2	Ilioinguinal iliohypogastric nerves block
	3	Transversus abdominis plane (TAP) block
	4	Midaxillary transversus abdominis plane block
	5	Subcostal transversus abdominis plane block
	6	Anterior quadratus lumborum block (QLB)
	7	Lateral quadratus lumborum block (QLB)
	8	Posterior quadratus lumborum block (QLB)
	9	Transversalis fascia plane (TFP) block
	10	Rhomboid intercostal plane block
Paraspinal	11	Paravertebral block (PVB)
	12	Intertransverse process (ITP) block
	13	Erector spinae plane (ESP) block
	14	Retrolaminar block (RLB)
Chest wall	15	Superficial serratus anterior plane (SAP) block
	16	Deep serratus anterior plane block (SAP)
	17	Superficial parasternal intercostal plane (PIP) block
	18	Deep parasternal intercostal plane (PIP) block
	19	Interpectoral plane (IPP) block
	20	Pectoserratus plane (PSP) block

## Data Availability

Not applicable.

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
