# Peer review of "Erector Spinae Plane Block and Chronic Pain: An Updated Review and Possible Future Directions"

_biology, 2023, doi:10.3390/biology12081073_

Round 1
Reviewer 1 Report
The present manuscript reviews “Erector Spinae Plane block and chronic pain: an updated review of the actual knowledge and a peek on the possible future directions”.
The ESPB is relatively new and unstudied, and the relationship of Erector Spinae Plane block (ESPB) and chronic pain is unclear. In this review, authors review the Erector Spinae Plane block rationale, mechanism of action, possible complication and discuss its use for chronic pain. However, there are still no randomized studies with high evidence on the use of ESP block in various types of chronic pain. The article is so long, and the focus, the relationship of Erector Spinae Plane block (ESPB) and chronic pain, is also not prominent. Therefore, the review can provide little useful information. And it is recommended to simplify and quickly enter into the subject of ESPB. Moreover, there is also some re-working needed in terms of grammar, style, organization, and methodological details.
I have the other following comments:
1: In the line 213 and 259, the Figure 1 (Graphic representation of ESP block) and Figure 2 (Coronal view of magnetic resonance imaging of two patients receiving ESP block with a 30 mL volume (gadolinium).The ESP block landmark is supposed to be added by the author with high resolution.
2: In the part of “ESP block rate of complications ”, line 292 to 294, the author described the “The most serious complication of any local anesthetic technique is systemic toxicity (LAST), which can occur if the anesthetic solution is inadvertently injected into a blood vessel or if too much solution is injected”. Recent research revealed that a female patient's local anesthetic plasma concentration was in excess of the minimum level after receiving 15ml of 0.33% ropivacaine on each side. (Huang, et al. Br J Anaesth. 2022;129(3):445-453 ). Furthermore, a study also demonstrated that female patients can absorb more local anesthetics in ESP (De Cassai A,et al 2021;46(1):86-89). What's more, a potential analgesic benefit of ESP is the continuous adsorption of local anesthetic into the blood. The author needs to take into account how ESP patients differ in terms of their age, gender, and other physiology factors.
Furthermore, previous research discovered that ESP has a broad compartment to absorb the local anesthetics, creating a para-spinal analgesic effect. The author is supposed to focus on latest studies in ESP. (De Cassai A,et al 2021;46(1):86-89; Black ND, Stecco C, et al 2021;132(3):899-905).
3: The title of this review is about chronic pain management of ESPB, the author should focus more on the “ chronic pain”. Furthermore, it may be more perspective that compared the analgesia effect of ESPB compared with other regional blocks in various surgeries.(Moorthy A, et al. Br J Anaesth. 2023;130(1):e137-e147;Pawa A, King C, et al. Br J Anaesth. 2023;S0007-0912(23)00019-3).
Author Response
Q1: In the line 213 and 259, the Figure 1 (Graphic representation of ESP block) and Figure 2 (Coronal view of magnetic resonance imaging of two patients receiving ESP block with a 30 mL volume (gadolinium).The ESP block landmark is supposed to be added by the author with high resolution.
A1: We added a new figure (New Figure 2) with sonoanatomy of ESP block
2: In the part of “ESP block rate of complications ”, line 292 to 294, the author described the “The most serious complication of any local anesthetic technique is systemic toxicity (LAST), which can occur if the anesthetic solution is inadvertently injected into a blood vessel or if too much solution is injected”. Recent research revealed that a female patient's local anesthetic plasma concentration was in excess of the minimum level after receiving 15ml of 0.33% ropivacaine on each side. (Huang, et al. Br J Anaesth. 2022;129(3):445-453 ). Furthermore, a study also demonstrated that female patients can absorb more local anesthetics in ESP (De Cassai A,et al 2021;46(1):86-89). What's more, a potential analgesic benefit of ESP is the continuous adsorption of local anesthetic into the blood. The author needs to take into account how ESP patients differ in terms of their age, gender, and other physiology factors.Furthermore, previous research discovered that ESP has a broad compartment to absorb the local anesthetics, creating a para-spinal analgesic effect. The author is supposed to focus on latest studies in ESP. (De Cassai A,et al 2021;46(1):86-89; Black ND, Stecco C, et al 2021;132(3):899-905).
Q2: We added the following:
"The most serious complication of any local anesthetic technique is systemic toxicity (LAST), which can occur if the anesthetic solution is inadvertently injected into a blood vessel or if too much solution is injected. Additionally, practitioners should be aware that the rate of adsorption of local anesthetics from the ESP plane into systemic circulation is relatively fast with Tmax happening at only five minutes after the injection. These findings have been consistently found for both lidocaine [25] and ropivacaine [31]. Until further data will be available physicians should take into account that different rates of adsorption could occur in different group of patients (in example related to age, gender, weight and so on)."
Q3: The title of this review is about chronic pain management of ESPB, the author should focus more on the “ chronic pain”. Furthermore, it may be more perspective that compared the analgesia effect of ESPB compared with other regional blocks in various surgeries.(Moorthy A, et al. Br J Anaesth. 2023;130(1):e137-e147;Pawa A, King C, et al. Br J Anaesth. 2023;S0007-0912(23)00019-3).
A3:We decided to not expand more the paragraph regarding other regional block to not loose the focus of the manuscript. However, we are available to furhter expand the text if editor or reviewer believe it is foundamental
Reviewer 2 Report
Thank you for permitting me to review this manuscript
the chapter in chronic pain is too long and need to be shortened as this review is supposed to focus on the effect of the erector spinae block to chronic pain , I understand that sometiomes there is need to fullfill the minimum words required but in this case it sounds better to focus on the block rather than chronic pain
Regional anesthesia is very helpfull in patients with chronic pain having surgery , this should be referenced and discussed
Table may be coupled with a figure representy the whole body and specifying the localization of each block
Figure 1 is fine and need to be coupled with an ultrasound pictures including with the spreading of the local anesthetics
Products with dosages may be usefull
The authirs should also cite other works in which other blocks had a positive or negative role in chronic pain
Author Response
Q1)the chapter in chronic pain is too long and need to be shortened as this review is supposed to focus on the effect of the erector spinae block to chronic pain , I understand that sometiomes there is need to fullfill the minimum words required but in this case it sounds better to focus on the block rather than chronic pain
A1) We reduced the lenght of chronic pain chapter and extend the esp block and chronic pain section
Q2)-Regional anesthesia is very helpfull in patients with chronic pain having surgery , this should be referenced and discussed
-The authirs should also cite other works in which other blocks had a positive or negative role in chronic pain
A2) We rewrote and expanded the esp block and chronic pain section
Q3)Table may be coupled with a figure representy the whole body and specifying the localization of each block
A3) we thanks the reviewer for the suggestion, but we believe that a table is more informative than an image, however to procude an image if editor or reviewer believe it is foundamental
Q4)Figure 1 is fine and need to be coupled with an ultrasound pictures including with the spreading of the local anesthetics
A4)Done
Q5)Products with dosages may be usefull
A5)We thanks the reviewer for his comment, however, there is a so much variety in both products and dosages that a table would be at least misleading.
Reviewer 3 Report
I read this manuscript with great interest. ESP block gained popularity after its description and many studies have been conducted. Thiswell written manuscript summarizes its use in chronic pain, and I have no additional comments.
Author Response
We thanks the reviewer for his kind comments
Reviewer 4 Report
I thoroughly enjoyed reading this well written and organized review of an emerging technique in the management of both acute and chronic pain. The manuscript does require some English editing, although the overall quality is quite good. I have two suggestions for the authors to consider: 1) a more in depth review of existing data regarding efficacy in acute and chronic pain, including directly addressing the quality of current evidence. 2) inclusion of US images of the ESP block.
Author Response
Q1) a more in depth review of existing data regarding efficacy in acute and chronic pain, including directly addressing the quality of current evidence.
A1: We expanded the esp block and esp block and chronic pain sections.
2) inclusion of US images of the ESP block.
Q2: Added
Round 2
Reviewer 1 Report
In the revised manuscript, the most important question did not be solved. The manuscript could provide little useful information about ESPB for chronic pain. And the revised manuscript did not include a detailed point-by-point explanation of how the authors responded to each of the points raised by the assessors. For example, "In lines 213 and 259, Figure 1 (Graphic representation of ESP block) and Figure 2 (Coronal view of magnetic resonance imaging of two patients receiving ESP block with a 30 mL volume (gadolinium). The ESP block landmark is supposed to be added by the author with high resolution."
Author Response
Dear Reviewer, in fact, as replied in previous round of reviews ESP block landmark could not be added in Figure 2 as it is a coronoral view of an ESP block, meaning that all the picture is the Erector spinae plane.
Regarding Figure 1 authors believe that the picture is clear as there is the needle with the tip in the ESP and the local anesthetic spreading from the tip. We do not believe that adding an * or any other landmark would enhance image readibility for the readers.
However, if Academic Editor believe it is appropriate we could add it.
Moreover, we have added an ultrasound image to help readers not familiar with ESP block to enhance cleariness on the technique
Reviewer 2 Report
the authors have successfully responded to my request or suggestion
Author Response
We would like to thanks the reviewer for his comments